# LLM Generated Persona is a Promise with a Catch

## Abstract

The use of large language models (LLMs) to simulate human behavior has gained significant attention, particularly through personas that approximate individual characteristics. Persona-based simulations hold promise for transforming disciplines that rely on population-level feedback, including social science, economic analysis, marketing research, and business operations. Traditional methods to collect realistic persona data face significant challenges: they are prohibitively expensive and logistically challenging due to privacy constraints, and often fail to capture multi-dimensional attributes, particularly subjective qualities. Consequently, synthetic persona generation with LLMs offers a scalable, cost-effective alternative. However, current approaches rely on ad hoc and heuristic generation techniques that do not guarantee methodological rigor or simulation precision, resulting in systematic biases in downstream tasks. Through extensive large-scale experiments including presidential election forecasts and general opinion surveys of the U.S. population, we reveal that these biases can lead to significant deviations from real-world outcomes. Based on the experimental results, we argue that **a rigorous and systematic science of persona generation is needed to ensure the reliability of LLM-driven simulations of human behavior.** We call for not only methodological innovations and empirical foundations but also interdisciplinary organizational and institutional support for the development of this field. To support further research and development in this area, we have open-sourced approximately one million generated personas, available for public access and analysis.

## 1 Introduction

Recent advances in LLM-driven simulations have enabled the creation of synthetic agents that approximate real-world populations' behaviors at scale, potentially transforming various fields such as social science (Manning et al., 2024), political science (Argyle et al., 2023), economics (Horton, 2023), marketing research (Sarstedt et al., 2024), clinical psychology (Wang et al., 2024), entertainment (Shao et al., 2023), personalization (Zollo et al.), and business applications such as recommendation systems and web testing (Tseng et al., 2024).

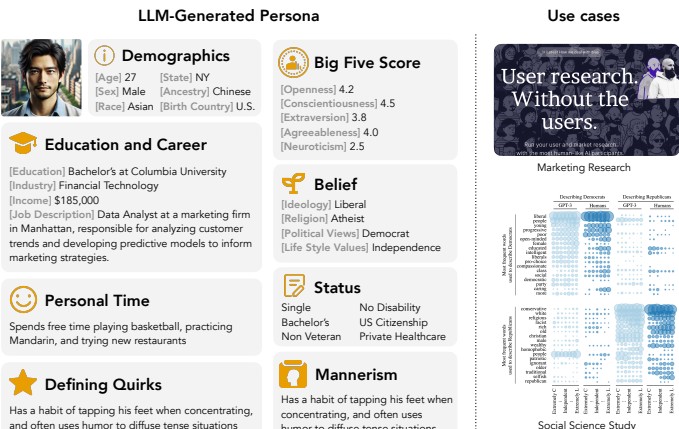

Figure 1: Left: An example of a LLM generated persona. Right: Applications of personas in the real world.

A central component of these simulations is the use of personas as shown in Figure 1: digital representations of individuals characterized by demographic, psychographic, and behavioral attributes. When provided with persona information, LLMs can generate responses

and simulate decision-making processes that aim to reflect those of real individuals (Park et al., 2024), serving as "silicon samples" to complement "human samples" for opinion testing (Argyle et al., 2023).

While "silicon samples" hold the potential to revolutionize societal-scale experimentation, a critical challenge remains: the systematic and reliable generation of persona profiles for targeted populations. Although large-scale collection of human data can maximize downstream simulation fidelity (Park et al., 2024), gathering such comprehensive data is far from trivial. It requires resource-intensive surveys and coordinated efforts that demand substantial financial investment, time, and operational support, while adhering to strict privacy regulations and ethical guidelines. On the other hand, existing large-scale real-world datasets (e.g., the U.S. Census (U.S. Census Bureau, 2024)) primarily contain marginal demographic information without capturing the joint distribution of multi-dimensional attributes. This fragmentation makes it impossible to reconstruct realistic, integrated personas that accurately mirror the complexity of real-world populations. Furthermore, these datasets often omit subjective attributes, such as lifestyle preferences or nuanced belief systems, due to privacy concerns or data collection limitations, despite their critical role in shaping individual opinions and values.

Meanwhile, LLM itself presents a viable solution to the above challenges by generating persona profiles directly in a cost-effective, efficient, and seemingly realistic manner. This nascent direction has received much attention recently from both academia and industry (Fröhling et al., 2024; Castricato et al., 2024; Ge et al., 2024; Schuller et al., 2024; Synthetic Users; CivicSync), where LLM-generated personas have been used to conduct surveys, marketing research, or even societal-scale simulations. Prior work has primarily focused on methodologies for scaling up the number of diverse personas, but there are no rigorous evaluations of their performance in different downstream applications or whether they faithfully capture a specific population's opinion at scale.

Given the huge potential and interests of LLM-based persona generation and the lack of a scientific community that studies this problem, we argue that **a science of persona generation needs to be developed to fully realize the potential of LLM persona simulation.** Specifically, we observe that the current scalable persona generating methods are significantly biased and non-representative of the real-world distribution. *One example is that in a 2024 presidential election simulation, results generated by a specific type of LLM-based synthetic personas predict a Democratic sweep across all U.S. states* (see Figure 2). Since these generated personas can be widely used in applications ranging from opinion simulation to product testing, *their inherent biases can lead to harmful consequences, including skewed public decision-making, reinforcement of discrimination and stereotypes, and potential harm to minority groups.*

While collecting better persona data is vital and nontrivial, we believe that both data collection and persona generation and simulation methods are complementary rather than mutually exclusive. Regardless of the dataset's size, generation methods are necessary to capture the complex, subjective aspects that traditional surveys often miss.

Therefore, we see an urgent need and call for a concerted effort from the AI community and interdisciplinary collaborations across social sciences to develop reliable methodologies, robust large-scale benchmarks, and broad community support to establish persona generation and silicon-sample simulation as a rigorous field. This paper is organized as follows:

1. *Systematization of Persona Generation Methods.* In Section 2, we categorize and systematize existing persona generation approaches, as summarized in Figure 3. This systematization lays the foundation for subsequent experimental evaluation and provides a structured framework for advancing the methodological rigor of persona generation.

2. *Current Persona Generation Practice Falls Short:* In Section 3, we present an extensive experimental evaluation of current persona generation practices. Using three commonly utilized persona types, we generate around 1,000,000 personas with six open-source LLMs and assess them across 500+ questions spanning diverse domains. The results demonstrate a substantial bias, amplified by the LLM-generated persona content. These findings critically undermine the validity of using LLMs as "silicon samples" and raise serious concerns about

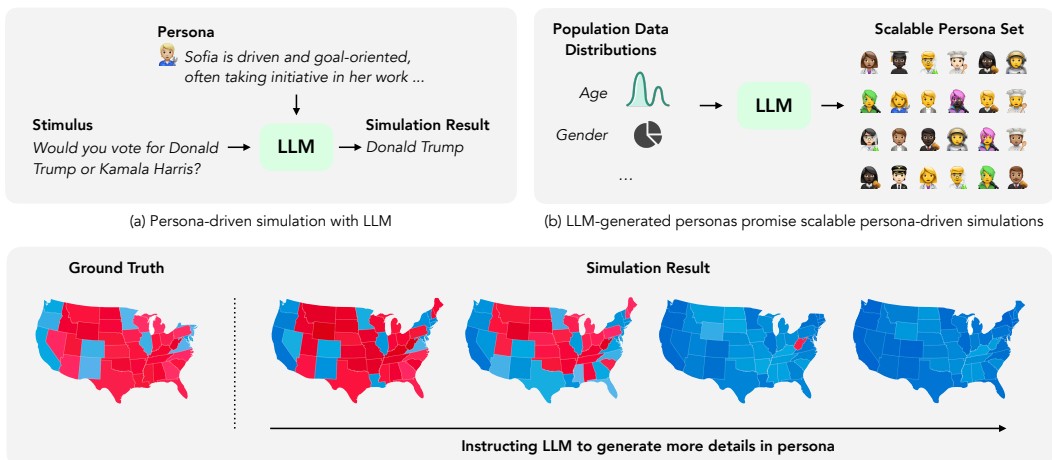

(a) Persona-driven simulation with LLM

(b) LLM-generated personas promise scalable persona-driven simulations

(c) We found that introducing more LLM-generated content into a persona's attributes exacerbates skew in simulated opinions and preferences

Figure 2: (a) Persona-driven simulation enables simulating human behaviors with LLMs. (b) Using LLM to generate personas promises scalable simulation of diverse population's behaviors. (c) We caution that improper usage of LLMs as persona generators may lead to homogeneous results.

applying LLM-generated personas in social science research and business decision-making without proper caution. Section C provides a semantic analysis of the LLM-generated persona profiles.

3. *A Path Towards Rigorous Persona Generation:* In Section 4, we outline key directions for establishing a more scientific and systematic approach to persona generation, including: (i) developing refined frameworks for identifying essential persona attributes, (ii) building theoretical and empirical foundations for persona profile calibration, (iii) creating large-scale datasets and benchmarks to validate and refine persona generation methods, and (iv) advocating for interdisciplinary research and careful application-oriented consideration of risks and biases in persona generation and simulation.

Overall, we advocate for increased attention from the broader research community to realize the transformative potential of widespread LLM silicon sample adoption, enabling rapid, cost-effective, safe, and human-centric tests and innovation. As a first step toward this goal, we open-source all generated personas in our experiments to support and accelerate community efforts in this direction.

**A study of LLM persona generation bias rather than traditional LLM bias.** While biases in LLMs have been extensively studied in the literature (Gallegos et al., 2024; Ferrara, 2023; Feng et al., 2023; Parrish et al., 2021; Bai et al., 2024; Lu et al., 2020; Li et al., 2020a), most prior work focuses on biases arising from *LLM simulation*. In contrast, we highlight a striking and underexplored source of bias introduced by *LLM persona generation* itself. Specifically, prior work in LLM bias has largely taken the persona generation process for granted, focusing instead on simulation outputs. In contrast, our study explicitly dissects the persona generation step, revealing that even if the simulation appears unbiased, the underlying generated personas may carry significant, unexamined biases that undermine their validity. Our findings indicate that this area requires significantly more attention to ensure fairness and representativeness in generated personas. For a comprehensive discussion of related work, please refer to Appendix A.

## 2 Persona-Driven Opinion Simulation

To highlight the shortcomings of current persona generation methods, we first establish the foundation for large-scale simulation experiments by implementing various commonly used

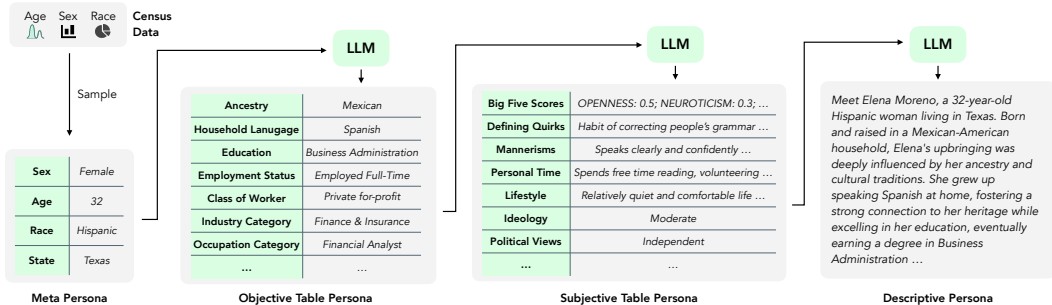

Figure 3: We categorize existing persona generation approaches into four tiers. Each tier adds more information generated by LLMs in generated personas to the previous tier.

persona generation strategies and simulation methods, drawing from both prior literature and community practices.

## 2.1 Persona Generation

We build on prior literature to construct three distinct types of personas: **Meta Personas**, **Tabular Personas**, and **Descriptive Personas**.

**Meta Persona.** Census data provides a foundational source for generating realistic personas due to its rich demographic information that reflects real-world population distributions. However, census data typically presents marginal distributions (e.g., percentages of different age groups and genders in New York), which limits its capacity for capturing joint probabilities. This limitation can lead to incongruous combinations of attributes (e.g., a person earning $500k annually but unable to afford health insurance). To address this, we adopt methods from Chang et al. (2024) and Argyle et al. (2023), leveraging data with joint distributions over key attributes, such as Age, Sex, Race, and State, provided by the U.S. Census Bureau. We term these structured, sampled personas **Meta Personas** (No LLM is involved in generating them). While these personas may lack diversity, they serve as a robust foundation for generating more varied and realistic personas.

**Tabular Personas.** To enhance diversity and better capture nuanced differences for opinion simulation, we extend the information encoded in **Meta Personas** using structured templates. Following the proposed method of Castricato et al. (2024), we employ LLMs to fill in additional attributes within these templates, conditioned on the generated meta personas. We propose two variants:

1. *Objective Tabular Personas*: These personas expand **Meta Personas** by adding attributes that can be objectively measured (e.g., Income, Education Level, Occupation, and Industry). We align these entries with the Census Bureau's predefined categories (detailed in the appendix), which ensures consistency and mitigates bias. For example, occupations are drawn from five predefined categories rather than generated freely by the LLM.

2. *Subjective Tabular Personas*: These personas extend *Objective Tabular Personas* by adding subjective or less standardized attributes (e.g., Political Affiliation, Leisure Preferences). Since these attributes lack comprehensive predefined categories, we prompt the LLM to generate open-ended responses, guided by templates designed to maximize diversity while preserving realism.

**Descriptive Personas.** The most flexible approach, **Descriptive Personas**, involves directly prompting the LLM to create freeform personas conditioned on **Meta Personas**. This method allows the generation of highly detailed and unconstrained personas by asking the LLM to expand beyond structured attributes into narrative-style descriptions. While this approach provides maximum freedom, it requires careful prompting and post-processing to ensure the plausibility and coherence of the generated personas.

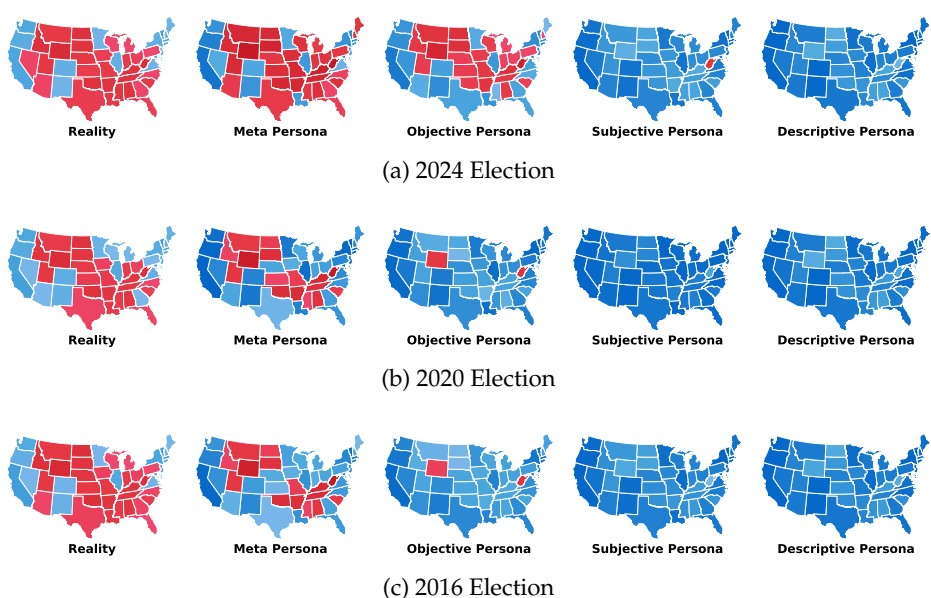

(a) 2024 Election

(b) 2020 Election

(c) 2016 Election

Figure 4: Persona-based simulations of elections 2016, 2020, and 2024.

Progressing from **Meta Personas** to **Tabular Personas** to **Descriptive Personas** entails a growing content generated by LLMs for diversity. This means that the persona types from left to right in Figure 3 gradually introduce richer, more varied attributes and open-ended descriptions, which theoretically can yield broader and more diverse opinion distributions. However, as we show later, **increasing the amount of LLM-generated persona content can exacerbate biases in the simulation results in various domains.**

To approximate opinion distributions and capture diversity at the U.S. national level, we first sample 1,000 personas per state using the corresponding joint distribution. We then generate 1,000 augmented personas for each of the persona types (**Meta**, **Tabular**, and **Descriptive**). During the opinion simulation, we present the language model with a short survey: a question accompanied by multiple choices, a persona description, and a template prompting the model to select the most suitable response based on the persona's attributes. We explore several prompt variants that encourage neutrality and mitigate unwanted biases or stereotypes, but observe no substantial performance differences across these variants. The specific prompts and templates can be found in Appendix D

### 2.2 Simulating Opinions with LLMs

We treat LLMs as impartial world models and simulators. Building on recent work such as Park et al. (2024), we hypothesize that, when conditioned on realistic and adequate information, LLMs can truthfully simulate a persona's opinion, effectively serving as digital twins. To simulate each persona's opinion, we first provide high-level instructions guiding the LLM to remain unbiased and objective while faithfully reflecting the persona's attributes. Next, we input a persona along with a multiple-choice question, instructing the LLM to directly select one of the provided answers. The final prompt, includes the simulation instruction, persona description, and the simulation question, is fed once into the language model. Additional details can be found in Appendix B.

## 3 Can Personas Simulate the Society?

We now present empirical findings from two main simulation scenarios: (1) U.S. elections from 2016, 2020, and 2024, and (2) general opinion surveys drawn from OpinionQA (Santurkar et al., 2023). Our analysis focuses on how relying on different persona types (with varying amounts of LLM-generated content) affects alignment with real-world data.

## 3.1 A Case Study with US Presidential Election Voting

Motivated by prior work in political and social opinion simulations Argyle et al. (2023), we explore how our persona-based approach can serve as a viable methodology for modeling society-level electoral outcomes. Given that most large language models (LLMs) are trained on data collected prior to the 2024 U.S. election, we first test on the 2024 presidential election to avoid potential confounds from post-2024 information.

We plot the map of each state in blue and red with different color scales to indicate the support rate. In Figure 4a, we show one particular experimentation where the persona model and the simulation model are both Llama 3.1 70B. Our results reveal that with increasing LLM-generated persona attributes, the simulation results tend to deviate more from real-world outcomes. Specifically, **Meta Personas** yield simulated voting distributions closest to reality data, whereas **Generative Personas** produce results most divergent from reality. Moreover, we observe that as the level of LLM-generated persona information rises, the simulated electorate shifts progressively toward left-leaning stances, ultimately culminating in a surprising scenario where every state appears to vote for the Democratic candidate.

To investigate whether historical election data—presumably included in the training sets—reduces model bias, we conduct similar simulations for the 2016 and 2020 election cycles. One might expect that since these events are part of the training data, the models would "remember" historical facts and produce more accurate electoral distributions. Contrary to this expectation, we observe a similar pattern of leftward drift, suggesting that awareness of past elections alone does not sufficiently counterbalance the influence of LLM-generated content.

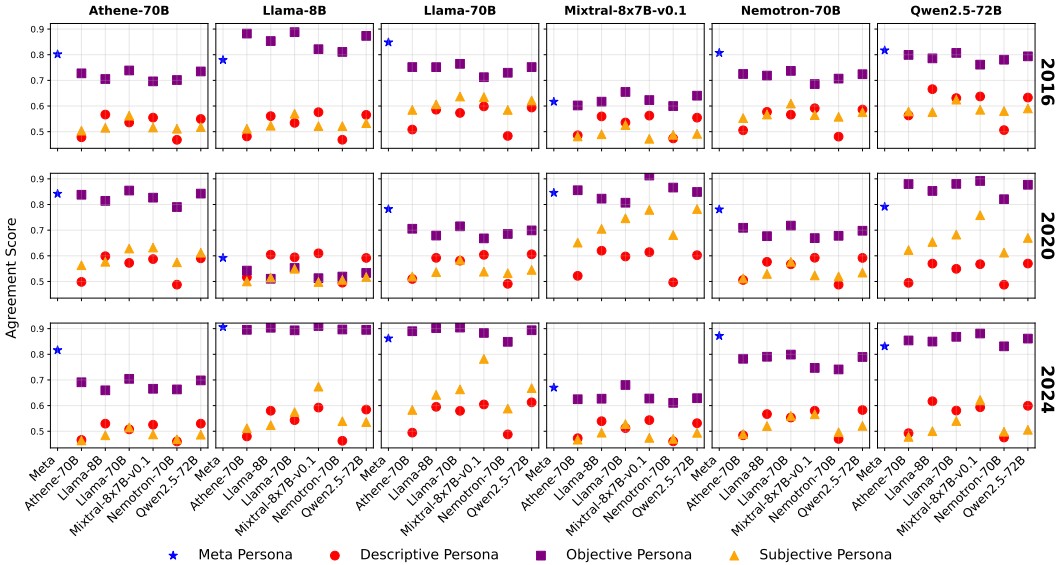

Figure 5: Alignment scores for cross-model simulation. Each column represents a simulation model, while the x-axis within each column corresponds to the persona generation model. The "meta" point is singular as it relies on sampling rather than generation.

**Quantitative evaluation.** To more rigorously compare our simulated results with ground truth, we adopt the alignment metric from Santurkar et al. (2023), using $1 - W(\hat{p}, p)$ as the alignment score. Here, $W(\hat{p}, p)$ denotes the Wasserstein distance between the simulated support rate ($\hat{p}$) and the actual voting rate ($p$), with higher alignment scores indicating results closer to real-world outcomes. Additionally, we perform *cross-simulation*, in which each LLM-generated persona set is simulated across all other language models. Our comprehensive assessment Figure 5 shows that this type of bias is not model-specific but rather universal in persona-driven simulations.

## 3.2 Exploring Bias Across Domains

We extended our analysis to assess whether the observed biases are limited to the political domain or span broader contexts.

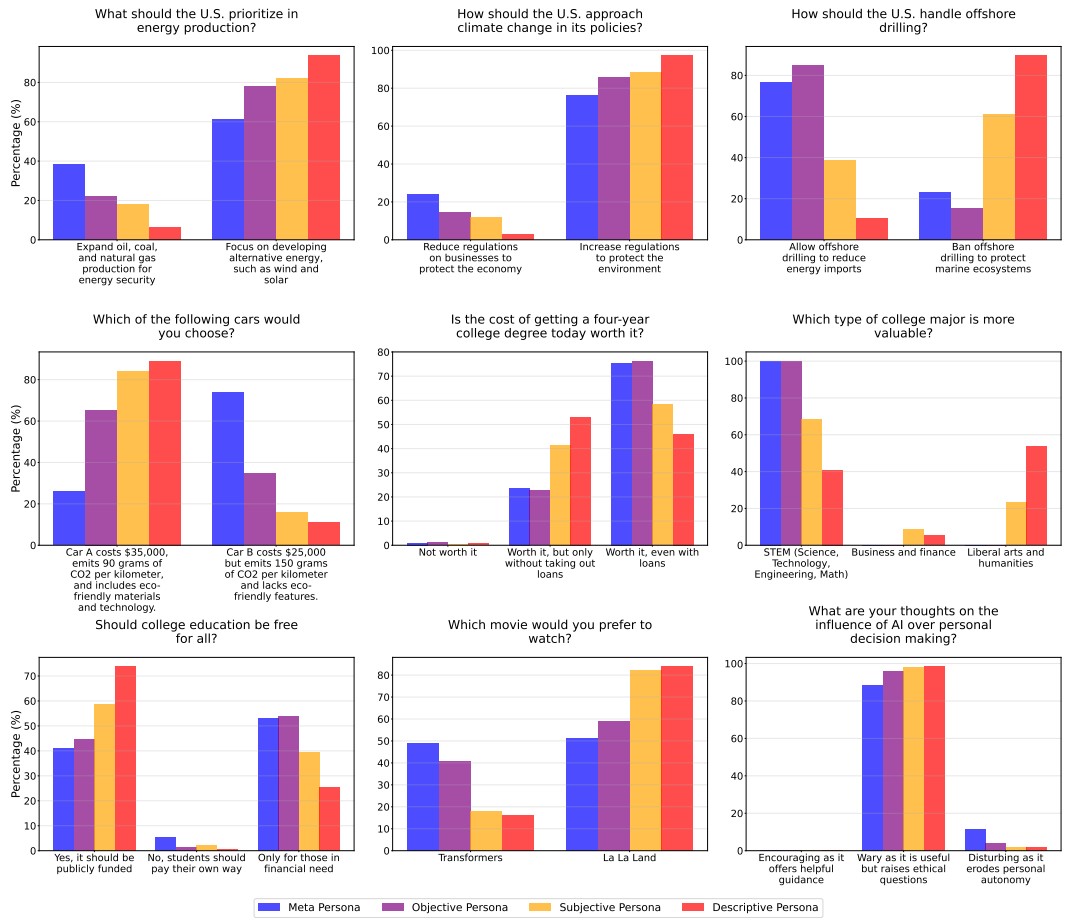

Figure 6: A preview of some specific topics that incurs interesting phenomenons

To this end, we designed 9 questions in Figure 6 covering five distinct domains: climate, consumer choices, education, entertainment, and technology.

For these questions, we do not have ground truth responses to measure alignment scores as in the political example. However, the plots provide a qualitative understanding of the general trend across different persona types' perspectives. Specifically, as persona types become more LLM-generated—from Meta to Descriptive—their perspectives shift from traditional to more progressive views. For example, LLM-generated personas increasingly *prefer environmentally friendly cars over cheaper options, value liberal arts over STEM majors, and choose movies like "La La Land" over "Transformers."*

While our experiments are not exhaustive, they reveal significant trends in how different persona models respond to diverse societal and individual decision-making scenarios. This raises concerns about deploying these "silicon samples" for social science and marketing research. *For instance, the hidden preferences of silicon samples, such as favoring electric cars over gas-powered vehicles, could bias the decision-making of a car business owner. Similarly, an entertainment company might be misled for the market trend if these samples show a bias for musical or romantic movies over action films.*

**OpinionQA.** To gain a more quantitative understanding, we evaluate various personas on the OpinionQA dataset (Santurkar et al., 2023), which provides a comprehensive evaluation of model behavior across a wide range of sensitive and broader topics.

The OpinionQA dataset collects questions and real US population's responses from Pew research surveys, covering a diverse range of topics, including sensitive areas such as race, misinformation, and gender, as well as broader subjects like science, biomedical research, and automation. This allows us to examine how different persona generation methods influence simulated opinions on various social, economic, and personal issues. We use the first 500 questions in OpinionQA, covering 15 distinct topics (see Figure 7 for the list of topics). For each question, we simulate responses using each of the four persona types and calculate the alignment score between the simulated distribution of answers and the ground truth distribution provided in the OpinionQA dataset. We can clearly see that persona opinions diverge more on controversial topics.

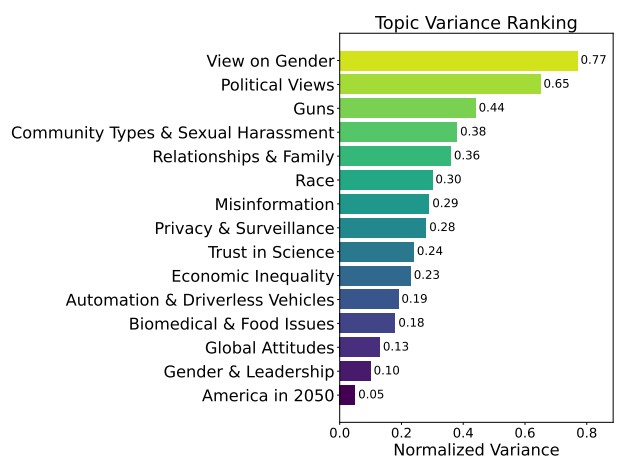

Figure 7: Rank of variance in alignment score among four persona types for the first 500 questions in Opinion QA based on topics.

## 4 Paths Forward: Toward a Scientific Approach to Persona

Our investigation reveals a critical limitation of current persona generation practices: none of the existing methods reliably produce realistic societal-level opinions. This underscores the urgent need for a rigorous and scientific approach to persona generation, one that moves beyond ad-hoc techniques and is borne out of principled methodologies. We put an initial analysis of persona in Appendix C. We further list a few potential directions and challenges that the community could work on.

**Identifying essential information needed in a persona.** A foundational challenge in persona-based simulation is identifying the essential information required for effective persona generation and how that information should be represented. The goal is to move beyond simply listing attributes to understanding what truly drives realistic simulation outcomes. Existing research offers conflicting evidence. While Argyle et al. (2023); Park et al. (2024); Salewski et al. (2023); Toubia et al. (2025) demonstrated that well-crafted conditioning can enable LLMs to simulate opinions aligned with real human responses, other studies such as Hu and Collier (2024); Gupta et al. (2023); Zheng et al. (2024); Beck et al. (2024) and our own experiments raise concerns about the efficacy and potential pitfalls of persona-based simulations. This discrepancy underscores the need to identify the crucial elements for effective persona-driven simulations, including which attributes are most important, such as demographic, psychographic (e.g., personality traits, values, attitudes, interests, lifestyles), behavioral (e.g., online activity), or contextual (e.g., social environment, current events), and the optimal format and prompting strategies for presenting that information to the LLM.

**Calibrating LLM-generated personas towards real population.** A parallel direction lies in accurately reconstructing realistic joint distributions of persona attributes from fragmented data sources, and subsequently calibrating these distributions to match a specific target population (Valliant et al., 2013). Even if we identify the crucial attributes for a given simulation, generating a population of personas requires sampling from the correct distributions. Existing datasets, such as the U.S. Census, often provide only marginal distributions of individual attributes (e.g., age, income, education level). This makes it impossible to sample from the true joint distribution. While Castricato et al. (2024) offers a first step by using

LLMs to filter out invalid attribute combinations sampled from marginal distributions, their method does not fully generalize to calibrate real-world joint distributions. Therefore, a crucial research direction is the development of robust sampling and calibration methods that can combine fragmented data and LLMs to accurately recover any target population.

**Open-source benchmark and datasets.** To accelerate progress in scientific understanding of the science of persona, we propose the creation of a large-scale, open-source benchmark dataset of rich and realistic persona profiles. Analogous to ImageNet Deng et al. (2009), a large-scale and open benchmark for persona generation would serve as a crucial resource for the research community. Specifically, this dataset would serve the following purposes: (i) a benchmark for evaluating the performance of different LLM-based persona generation methods; (ii) a training dataset such as (Toubia et al., 2025) for developing and testing new persona generation methods; and (iii) a high-quality profile library of diverse, realistic population-level personas suitable for direct use in "silicon sample" simulations. Constructing such a comprehensive dataset necessitates addressing data privacy concerns and requires a substantial investment of time and resources. However, we believe the potential benefits outweigh the effort.

**Interdisciplinary research and broad community collaborations.** Persona-driven simulation has the potential to revolutionize numerous fields, making it necessary for interdisciplinary collaboration between AI researchers and domain experts. Specifically, the ability to conduct rapid, cost-effective human experiments could be transformative for both academic fields, such as social science, economics, and political science (where human studies are crucial for advancement), and industry applications, like software and web design (where A/B testing is prevalent). We encourage broader exploration of silicon sample applications to better understand their potential and limitations, thereby advancing the field through collective community effort.

## 5 Discussions

While our paper highlights the current flaws and potential of LLM-generated personas for various applications, it is important to acknowledge that other approaches, such as personas derived from real-world data only, can achieve significantly higher prediction accuracy at the population level compared to synthetic ones. For example, Park et al. (2024) simulated over 1000 people based on their respective survey results and obtained 85% accuracy for prediction; similarly, Toubia et al. (2025) collected responses from over 2,000 participants across 500 questions, achieving 88% accuracy on held-out evaluation data. While we acknowledge the superior performance of real data, we have to point out that this approach fails to adequately address cost and ethical concerns. For instance, Park et al. (2024) could not release their dataset publicly, while Toubia et al. (2025) compensated thousands of participants with substantial payments, investing around $100,000 total. In addition to that, a variety of papers have supported the role of synthetic data in model training and improving simulation (Drechsler and Haensch, 2023; Eckman et al., 2024). Though not directly related to simulating human behaviors with LLMs, they provide foundational support for our position.

The limitations of real-world data further highlight the potential of synthetic personas, but only if they are constructed properly. This reinforces the need for a dedicated scientific community focused on understanding the capabilities and limitations of LLMs in persona generation and simulation, in order to fully unlock the potential of LLM-based persona modeling. This community must develop calibration methods for reducing biases, establish best practices for persona creation, and create benchmarks for assessing simulation results. Through interdisciplinary collaboration, we can envision a world where LLM-generated personas not only revolutionize various social science domains, but also transform our society with high-fidelity digital twins.

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

## A   Related Work

**Persona generation.**   Synthetic persona generation (Tseng et al., 2024; Chen et al., 2024) involves creating artificial user profiles for a variety of downstream tasks such as political simulations Lee et al. (2024) and economic research Horton (2023). Traditional approaches often rely solely on real-world datasets, such as census and survey data, to ensure statistical validity while addressing scalability and diversity challenges (Argyle et al., 2023; Chang et al., 2024; Huang et al., 2023; Xu et al., 2023). For example, agent-based modeling studies generate synthetic populations by sampling attributes (age, income, etc.)  from census or survey distributions, ensuring the aggregate matches real demographics Chapuis et al. (2022). Common techniques include clustering user data, factor analysis, and matrix decomposition to identify archetypal personas from large datasets Jansen et al. (2022). Generative personas, on the other hand, leverage LLMs to automatically scale up the number of distinct personas. Generative tabular personas attempt to balance demographic validity with enriched attributes by either employing LLMs (Castricato et al., 2024) or probabilistic methods (Li et al., 2020b) to fill in extra characteristics. Descriptive personas leverage LLMs to directly generate free-form personas conditioned on text input Ge et al. (2024). While lacking validity, descriptive personas serve as the most common approach for large-scale persona generation in the industry.

**LLM Driven Simulation.**   LLMs have enabled new possibilities in simulating human-like interactions and societal behaviors. Early works demonstrated feasibility in diverse settings, from digital twins of real individuals (Park et al., 2024; 2023) and role-playing of famous characters (Wang et al., 2023; Shao et al., 2023; Chen et al., 2024) or specific personalities (Caron and Srivastava, 2022), to society simulations (Argyle et al., 2023; Chuang et al., 2024; Yang et al., 2024; AL et al., 2024; Zhou et al., 2023; Pang et al., 2024).  However, though those works provide foundational support for our research, they primarily focus on demonstrating feasibility rather than conducting rigorous, large-scale simulations across diverse populations.  In our work, instead, we systematically examined how diverse personas influence the outcomes of LLM-driven simulations. Prior studies, such as (Park et al., 2024; Argyle et al., 2023), have shown that conditioning simulations on well-crafted personas can yield results that closely approximate real-world behavior.  However, while these works validate the feasibility of such simulations, none of them provide rigorous evidence that conditioning on the right persona guarantees accurate behavior. As an orthogonal direction, future research should rigorously refine simulation methodologies to improve both realism and scalability.

**LLM Bias.**   Bias in LLMs has been widely studied Gallegos et al. (2024); Ferrara (2023); Feng et al. (2023); Parrish et al. (2021); Bai et al. (2024); Lu et al. (2020); Li et al. (2020a); Blodgett et al. (2020); Sheng et al. (2021); Ma et al. (2024), including in LLM-driven simulations. Recent work shows that assigning personas to LLMs can expose hidden biases and harm reasoning Gupta et al. (2023), and that role-play increases both response quality and bias risk, especially with dynamic roles Zhao et al. (2024). In political simulations, LLMs overemphasize political homophily Chang et al. (2024) and favor English-speaking, democratic systems Qi et al. (2024). Distinct from prior work, our study investigates bias brought by persona generation, rather than the simulation itself.

## B   Additional Simulation Details

### B.1   Language Model Selection

We evaluate six open-source language models: *Athene 70B*, *Llama 3.1 8B*, *Llama 3.1 70B*, *Mistral-8x7B Instruct V0.1*, *Nemetron 70B*, and *Qwen 2.5B*. Our selection is driven by two key considerations:

1. **Alignment strategy.** Some models, such as the *Llama* family, are purely instruction-tuned, while others (e.g., *Athene* and *Nemetron*) undergo additional refinement through Reinforcement Learning from Human Feedback (RLHF).

2. **Geographic diversity.** We include models like *Qwen* and *Mistral* that are developed and trained primarily outside the United States, allowing us to investigate whether the training origin influences model biases or responses.

## B.2   A Special LLM

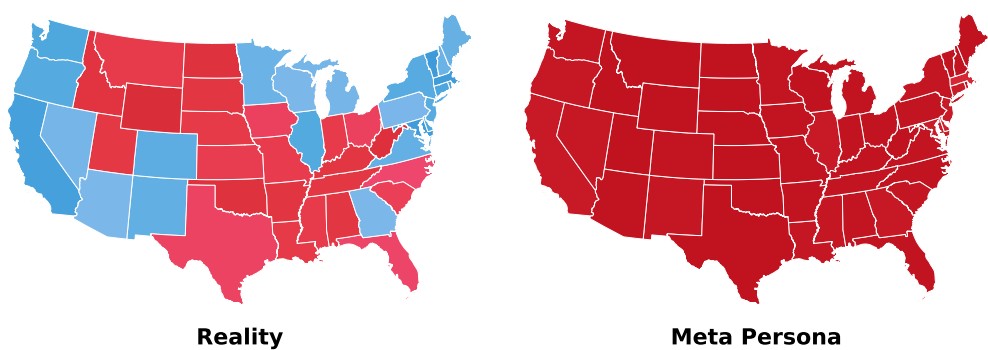

Figure 8: Yi-34B Chat Model Simulation of 2020 Election

Most language models exhibit a significant bias toward politically left-leaning perspectives. However, our analysis identified an exception in the Yi-34B chat model, which demonstrates a substantial bias toward politically right-leaning viewpoints. This discovery does not diminish our overall findings; instead, it reinforces the importance of carefully selecting and aligning language models for persona generation and simulation tasks.

## C   A Closer Look into Persona Profiles

What characteristics and biases emerge in the personas themselves as LLMs are given more generative freedom? Having established that increasing LLM-generated content leads to systematic biases in simulation outcomes, we now examine the inherent patterns in how these personas are constructed. We conducted sentiment analysis over generated persona using TextBlob (Loria, 2018). Our sentiment analysis in Figure 9a reveals that subjectivity increases as we instruct LLM to generate more details in persona. In addition, sentiment becomes more positive with more LLM-generated details, with descriptive persona showing significantly more positive sentiment polarity.

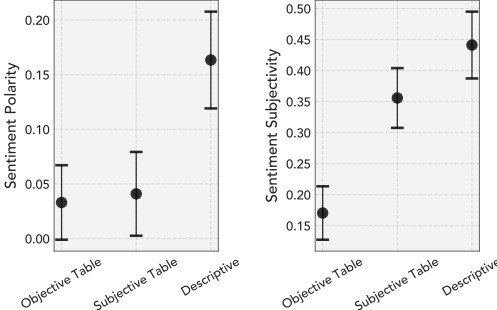

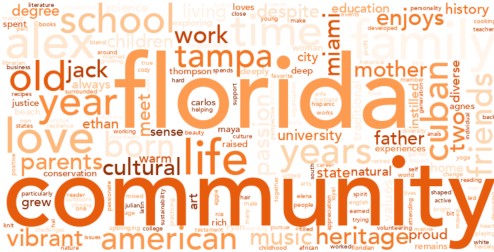

(a) Sentiment analysis of LLM-generated persona using TextBlob (Loria, 2018). As LLM injects more details in personas, sentiment polarity becomes more positive, and subjectivity increases.

(b) Word cloud of Florida descriptive persona.

Figure 9: Analysis of LLM-generated personas through sentiment and word distribution.

Qualitative word cloud analysis of the language patterns in descriptive personas supports these quantitative findings (Figure 9b). The prevalence of positively-valenced terms ("love", "proud") alongside community-oriented words ("family", "community") indicates LLMs systematically generate personas with optimistic life outlooks and strong social connections. Terms related to achievement and stability ("education", "work", "cultural", "heritage") further reinforce this positive framing. Notably absent are terms reflecting life challenges, social difficulties, or negative experiences, suggesting LLMs may be systematically avoiding less favorable characterizations. These findings help explain the previously observed biases in simulation outcomes: the combination of positive sentiment and highly emotional characterizations in the underlying personas naturally leads to skewed responses, particularly on social issues where emotional reasoning may play a stronger role.

## C.1 Feedback Collection

Instead of computing token-level log probabilities for each choice, we aggregate the counts of each selected choice across the simulated population, leveraging the instruction-following capabilities of modern LLMs.

# D Persona Generation and Simulation Prompts

## D.1 Persona Generation System Prompts

```
You are an AI assistant specialized in detailed and unbiased persona
    generation for opinion simulations. Your task is to generate a
    specific, realistic, and diverse persona based on the provided
    demographic information and fill in a comprehensive JSON template.
```

## D.2 Objective Tabular Persona

**Persona Generation Instructions**

```
### INSTRUCTIONS ###
1. You will be provided with a persona meta file that has the core
    demographic information of a person.
2. You will also be provided with a final persona template. Your task is
    to create a detailed, concrete persona that is fully consistent with
    ALL features in the given metadata by filling the template.
3. Elaborate on all metadata points, providing specific details that
    flesh out the persona while remaining true to the given information.
4. For all of the features in the metadata, you will be provided with a
    range of values in the VALUE RANGES AND CATEGORIES section below.
    Select one of the values for each of the features. DO NOT ADD EXTRA
    INFORMATION OR ELABORATION TO THE VALUES. DO NOT ADD EXTRA FEATURES
    TO THE TEMPLATE.
5. IMPORTANT: Place your entire response in the ### PERSONA GENERATION
    ### section below. Start your response with 'Persona:' and then
    provide only the persona description. Do not include any other
    prefixes, headers, or additional text.

### VALUE RANGES AND CATEGORIES ###

ANCESTRY: [
    "British",
    "Irish",
    "German",
    "Italian",
    "Polish",
    "French",
    "Norwegian",
    "Dutch",
```

```
632       "Swedish",
633       "Russian",
634       "Chinese",
635       "Filipino",
636       "Asian Indian",
637       "Vietnamese",
638       "Korean",
639       "Japanese",
640       "Mexican",
641       "Puerto Rican",
642       "Cuban",
643       "African American",
644       "West Indian",
645       "Arab",
646       "American Indian"
647   ]
648
649   HOUSEHOLD_LANGUAGE: [English, Spanish, Other Indo-European,
650       Asian/Pacific Islander languages, Other]
651
652   EDUCATION: [Less than HS, HS Graduate, Some College, Bachelor's,
653       Graduate Degree]
654
655   EMPLOYMENT_STATUS: [Employed, Unemployed, Not in Labor Force]
656
657   CLASS_OF_WORKER: [Private, Government, Self-employed, Unpaid family
658       worker]
659
660   INDUSTRY_CATEGORY: [
661       "Management, business, science, and arts occupations",
662       "Service occupations",
663       "Sales and office occupations",
664       "Natural resources, construction, and maintenance occupations",
665       "Production, transportation, and material moving occupations"
666   ]
667
668   OCCUPATION_CATEGORY: [
669       "Management, business, science, and arts occupations": [
670           "Management, business, and financial occupations",
671           "Computer, engineering, and science occupations",
672           "Education, legal, community service, arts, and media
673               occupations",
674           "Healthcare practitioner and technical occupations"
675       ],
676       "Service occupations": [
677           "Healthcare support occupations",
678           "Protective service occupations",
679           "Food preparation and serving related occupations",
680           "Building and grounds cleaning and maintenance occupations",
681           "Personal care and service occupations"
682       ],
683       "Sales and office occupations": [
684           "Sales and related occupations",
685           "Office and administrative support occupations"
686       ],
687       "Natural resources, construction, and maintenance occupations": [
688           "Farming, fishing, and forestry occupations",
689           "Construction and extraction occupations",
690           "Installation, maintenance, and repair occupations"
691       ],
692       "Production, transportation, and material moving occupations": [
693           "Production occupations",
694           "Transportation occupations",
695           "Material moving occupations"
696       ]
```

```
697  ]
698
699  INCOME: [Range $0-$1,000,000 annually]
700
701  MARITAL_STATUS: [Never Married, Married, Divorced, Widowed, Separated]
702
703  HOUSEHOLD_TYPE: [Family, Non-family]
704
705  PLACE_OF_BIRTH: [US State, Foreign Country]
706
707  VETERAN_STATUS: [Veteran, Non-veteran]
708
709  DISABILITY: [None, Physical, Mental, Both]
710
711  HEALTH_INSURANCE: [Private, Public, None]
712
713  ### RESPONSE FORMAT ###
714  Persona: [The completed FINAL PERSONA TEMPLATE]
715
716  ### PERSONA METADATA ###
717  {METADATA}
718
719  ### FINAL PERSONA TEMPLATE ###
720  {TEMPLATE}
721
722  ### PERSONA GENERATION ###
723
```

**Template**

```
726  "AGE": "",
727  "SEX": "",
728  "RACE": "",
729  "STATE": "",
730  "ANCESTRY": "",
731  "HOUSEHOLD_LANGUAGE": "",
732  "EDUCATION": "",
733  "EMPLOYMENT_STATUS": "",
734  "CLASS_OF_WORKER": "",
735  "INDUSTRY_CATEGORY": "",
736  "OCCUPATION_CATEGORY": "",
737  "INCOME": "",
738  "MARITAL_STATUS": "",
739  "HOUSEHOLD_TYPE": "",
740  "FAMILY_PRESENCE_AND_AGE": "",
741  "PLACE_OF_BIRTH": "",
742  "CITIZENSHIP": "",
743  "VETERAN_STATUS": "",
744  "DISABILITY": "",
745  "HEALTH_INSURANCE": ""
746
```

**D.3   Subjective Tabular Persona**

**Persona Generation Instructions**

```
750  ### INSTRUCTIONS ###
751  1. You will be provided with a persona meta file that has the core
752     demographic information of a person.
753  2. You will also be provided with a final persona template. Your task is
754     to create a detailed, concrete persona that is fully consistent with
755     ALL features in the given metadata by filling the template.
756  3. Elaborate on all metadata points, providing specific details that
757     flesh out the persona while remaining true to the given information.
758  4. For some of the features, you will be provided with a range of values
759     in the VALUE RANGES AND CATEGORIES section below. Select one of the
```

```
760        values for each of the features. DO NOT ADD EXTRA INFORMATION for
761        those features.
762   5. For the other features, fill in the values with a reasonable and
763        succinct description. Be as objective as possible.
764   6. IMPORTANT: Place your entire response in the ### PERSONA GENERATION
765        ### section below. Start your response with 'Persona:' and then
766        provide only the persona description. Do not include any other
767        prefixes, headers, or additional text.
768
769   ### VALUE RANGES AND CATEGORIES ###
770
771   ANCESTRY: [
772        "British",
773        "Irish",
774        "German",
775        "Italian",
776        "Polish",
777        "French",
778        "Norwegian",
779        "Dutch",
780        "Swedish",
781        "Russian",
782        "Chinese",
783        "Filipino",
784        "Asian Indian",
785        "Vietnamese",
786        "Korean",
787        "Japanese",
788        "Mexican",
789        "Puerto Rican",
790        "Cuban",
791        "African American",
792        "West Indian",
793        "Arab",
794        "American Indian"
795   ]
796
797   HOUSEHOLD_LANGUAGE: [English, Spanish, Other Indo-European,
798        Asian/Pacific Islander languages, Other]
799
800   EDUCATION: [Less than HS, HS Graduate, Some College, Bachelor's,
801        Graduate Degree]
802
803   EMPLOYMENT_STATUS: [Employed, Unemployed, Not in Labor Force]
804
805   CLASS_OF_WORKER: [Private, Government, Self-employed, Unpaid family
806        worker]
807
808   INDUSTRY_CATEGORY: [
809        "Management, business, science, and arts occupations",
810        "Service occupations",
811        "Sales and office occupations",
812        "Natural resources, construction, and maintenance occupations",
813        "Production, transportation, and material moving occupations"
814   ]
815
816   OCCUPATION_CATEGORY: [
817        "Management, business, science, and arts occupations": [
818            "Management, business, and financial occupations",
819            "Computer, engineering, and science occupations",
820            "Education, legal, community service, arts, and media
821                occupations",
822            "Healthcare practitioner and technical occupations"
823        ],
824        "Service occupations": [
```

```
825          "Healthcare support occupations",
826          "Protective service occupations",
827          "Food preparation and serving related occupations",
828          "Building and grounds cleaning and maintenance occupations",
829          "Personal care and service occupations"
830      ],
831      "Sales and office occupations": [
832          "Sales and related occupations",
833          "Office and administrative support occupations"
834      ],
835      "Natural resources, construction, and maintenance occupations": [
836          "Farming, fishing, and forestry occupations",
837          "Construction and extraction occupations",
838          "Installation, maintenance, and repair occupations"
839      ],
840      "Production, transportation, and material moving occupations": [
841          "Production occupations",
842          "Transportation occupations",
843          "Material moving occupations"
844      ]
845  ]
846
847  INCOME: [Range $0-$1,000,000 annually]
848
849  MARITAL_STATUS: [Never Married, Married, Divorced, Widowed, Separated]
850
851  HOUSEHOLD_TYPE: [Family, Non-family]
852
853  PLACE_OF_BIRTH: [US State, Foreign Country]
854
855  VETERAN_STATUS: [Veteran, Non-veteran]
856
857  DISABILITY: [None, Physical, Mental, Both]
858
859  HEALTH_INSURANCE: [Private, Public, None]
860
861  IDEOLOGY: [Very Liberal, Liberal, Moderate, Conservative, Very
862      Conservative]
863
864  POLITICAL_VIEWS: [Democrat, Republican, Independent, Other]
865
866  ### RESPONSE FORMAT ###
867  Persona: [The completed FINAL PERSONA TEMPLATE]
868
869  ### PERSONA METADATA ###
870  {METADATA}
871
872  ### FINAL PERSONA TEMPLATE ###
873  {TEMPLATE}
874
875  ### PERSONA GENERATION ###
876
```

**Template**

```
879  "AGE": "",
880  "SEX": "",
881  "RACE": "",
882  "STATE": "",
883  "ANCESTRY": "",
884  "HOUSEHOLD_LANGUAGE": "",
885  "EDUCATION": "",
886  "EMPLOYMENT_STATUS": "",
887  "CLASS_OF_WORKER": "",
888  "INDUSTRY_CATEGORY": "",
889  "OCCUPATION_CATEGORY": "",
```

```
890  "DETAILED_JOB_DESCRIPTION": "",
891  "INCOME": "",
892  "MARITAL_STATUS": "",
893  "HOUSEHOLD_TYPE": "",
894  "FAMILY_PRESENCE_AND_AGE": "",
895  "PLACE_OF_BIRTH": "",
896  "CITIZENSHIP": "",
897  "VETERAN_STATUS": "",
898  "DISABILITY": "",
899  "HEALTH_INSURANCE": "",
900  "BIG_FIVE_SCORES": {
901  "OPENNESS": "",
902  "CONSCIENTIOUSNESS": "",
903  "EXTRAVERSION": "",
904  "AGREEABLENESS": "",
905  "NEUROTICISM": ""
906  },
907  "DEFINING_QUIRKS": "",
908  "MANNERISMS": "",
909  "PERSONAL_TIME": "",
910  "LIFESTYLE": "",
911  "IDEOLOGY": "",
912  "POLITICAL_VIEWS": "",
913  "RELIGION": "",
914  "COGNITIVE_DIFFICULTY": "",
915  "ABILITY_TO_SPEAK_ENGLISH": "",
916  "VISION_DIFFICULTY": "",
917  "FERTILITY": "",
918  "HEARING_DIFFICULTY": ""
```

### D.4  Descriptive Persona

**Persona Generation Instructions**

```
### INSTRUCTIONS ###
1. You will be provided with a persona meta file that has the core
   demographic information of a person.
2. Your task is to create a detailed, diverse, and vivid persona that is
   fully consistent with ALL features in the given metadata.
3. Elaborate on all metadata points, providing specific details that
   flesh out the persona while remaining true to the given information.
4. For any ranges or categories provided in the metadata, select and
   specify exact values or details within those ranges/categories.
5. Ensure diversity in perspectives, backgrounds, and personality
   traits. Provide enough specific details to make the persona feel
   real and three-dimensional.
6. Maintain diversity by acknowledging various experiences within the
   demographic group, but commit to specific details for this
   individual persona.
7. IMPORTANT: Place your entire response in the ### PERSONA GENERATION
   ### section below. Start your response with 'Persona:' and then
   provide only the persona description. Do not include any other
   prefixes, headers, or additional text.

### RESPONSE FORMAT ###
Persona: [A detailed, vivid, and diverse description of a specific
   individual. Ensure all details are consistent with and elaborate
   upon the provided metadata.]

### PERSONA METADATA ###
{METADATA}

### PERSONA GENERATION ###
```

### D.5 Opinion Simulation Prompts

```
You are an AI assistant tasked with generating realistic opinions based
    on a given persona and a specific topic.

### TASK ###
You will simulate a persona answering a multiple-choice opinion
    question. Select the answer that best matches your persona's
    viewpoint and interests.

### GUIDELINES ###
1. Be Faithful to the Persona: Ensure your answer is consistent with the
    persona's data.
2. Focus on Relevant Aspects: Center your reasoning on the relevant
    factors that would influence the persona's opinion on that topic.
3. Be Objective: Avoid injecting personal bias or overly politically
    correct views that may not align with the persona's standpoint.

### INSTRUCTIONS ###
- Choose ONE option (A, B, C, or D depending on the number of options)
    that best fits the persona
- If multiple answers are possible, randomly select based on their
    probability
- Always pick an option, even in unclear cases - treat it as a
    forced-choice survey
- Output format: 'Answer: [Letter]' only, no explanation needed

### PERSONA ###
{PERSONA}

### QUESTION ###
{QUESTION}

### YOUR RESPONSE ###
```

