# OpenReview forum: "LLM Generated Persona is a Promise with a Catch"
_colmweb.org/COLM/2025/Workshop/Social_Sim — Social Sim'25_

### Official Review · Reviewer_FxWz · 2025-07-17

**Rating:** 6
**Overall Assessment:** 3
**Confidence:** 4

**Review:**

Please refer to the section 'Summary of strengths' and 'Summary of weakness' for the review.

**Comments Suggestions And Typos:**

Line 554: Nemetron -> Nemotron, Qwen 2.5B -> I guess some model in Qwen 2.5 family (probably 72B as in the main text) ?

**Ethical Concerns:**

No visible ethical concerns.

**Paper Summary:**

The authors examine the growing practice of employing LLMs to generate synthetic personas, which are subsequently used as "silicon samples" in opinion simulation tasks. They (1) propose a four-tier taxonomy: Meta, Objective-Tabular, Subjective-Tabular, and Descriptive, to categorize existing persona-generation techniques, (2) create ~ 1M personas using six LLMs, (3) conduct extensive simulations covering U.S. presidential elections and OpinionQA, and (4) demonstrate that increasing reliance on LLM-generated attributes leads to systematically left-leaning biases in simulated outcomes.. Persona generation itself may contain a previously under-appreciated source of bias and establishing a rigorous "science of persona generation" will be critical, including improved calibration techniques, standardized benchmarks, and interdisciplinary oversight.

**Relevance:**

5

**Summary Of Strengths:**

1. Systematization of methods. Figure 3’s tiered taxonomy summarizes a previously fragmented literature, clearly showing methods ranging from census-based sampling (without LLM) to free-form narrative generation.

2. Robust empirical analysis. The generation and evaluation of over 1M synthetic personas across more than 500 survey questions offer the most extensive quantitative investigation into persona-induced bias to date. (Figures 4 and 5)

3. Multi-domain validation. By supplementing electoral simulations with OpinionQA topics and custom-designed climate and consumer-related questions (Fig. 6), the paper convincingly demonstrates that biases extend beyond political contexts, although I would like to ask the authors the design process of 9 questions in Fig. 6.

4. Valuable resource contribution. The public release of  ~1M synthetic personas and accompanying codebase substantially lowers barriers to replication and facilitates future studies in the field.

**Summary Of Weaknesses:**

1. Missing mitigation baselines. While the study exposes bias, it does not test calibration or debiasing strategy (e.g., post‑stratification weighting, temperature/Top‑p sweeps, choice of model between post-trained and pre-trained base models, or even reasoning models).

- Also, there seems to be some other persona generation strategies that does not fall into the pre-defined 4 categories, like Concordia's demographic information conditioning (https://github.com/google-deepmind/concordia), Virtual Personas (https://arxiv.org/abs/2407.06576) backstory. I am curious how other persona generation strategies would behave.

2. During the introduction, authors present various venues LLM generated persona can be applied to. However, the main experiments are centered around the opinion prediction task. Would there be any other results on different setting, like in the consumer research setting how the bias are manifested? Current experiment scope seems to be too narrow compared to the argument in the introduction.

---

### Official Review · Reviewer_ZLs2 · 2025-07-21
**Systematic Bias in LLM-Generated Personas: Important Finding but Limited Solutions**

**Rating:** 6
**Overall Assessment:** 4
**Confidence:** 3

**Review:**

This paper addresses a critical issue as LLM personas gain industry adoption. The tiered experimental design effectively isolates LLM content effects, showing compelling evidence of systematic bias. However, the work remains diagnostic without explaining mechanisms or offering solutions. The US-centric evaluation limits global applicability. Missing statistical rigor, debiasing attempts, and cross-cultural validation. Important warning but incomplete as a research contribution.

**Comments Suggestions And Typos:**

Add statistical tests throughout
Test non-US contexts for fair evaluation
Explore prompt engineering for debiasing
Compare to human-written personas

**Paper Summary:**

This paper investigates biases in LLM-generated personas for behavioral simulations. Authors create four persona types with increasing LLM content: Meta (census-based), Objective Tabular, Subjective Tabular, and Descriptive. Testing on US elections and 500+ opinion questions reveals that more LLM-generated content leads to systematic liberal bias.

**Relevance:**

5

**Summary Of Strengths:**

Identifies critical real-world problem with industry impact
Clean experimental design isolating LLM content effects
Large-scale validation (1M personas, 6 models)
Clear evidence of systematic bias

**Summary Of Weaknesses:**

Limited to US context - unfair global generalization
Missing statistical significance tests
No comparison to alternative methods

---

### Meta-Review · Area_Chair_FKjX · 2025-07-21

**Recommendation:** Accept

**Metareview:**

--